# Influence of the Computer-Aided Static Navigation Technique on the Accuracy of the Orthodontic Micro-Screws Placement: An In Vitro Study

**DOI:** 10.3390/jcm10184127

**Published:** 2021-09-13

**Authors:** Paulina Rodríguez Torres, Sergio Toledano Gil, Álvaro Zubizarreta-Macho, María Bufalá Pérez, Elena Riad Deglow, Georgia Tzironi, Alberto Albaladejo Martínez, Sofía Hernández Montero

**Affiliations:** 1Department of Implant Surgery, Faculty of Health Sciences, Alfonso X el Sabio University, 28691 Madrid, Spain; prodrtor@uax.es (P.R.T.); stolegii@myuax.com (S.T.G.); mperebuf@uax.es (M.B.P.); elenariaddeglow@gmail.com (E.R.D.); shernmon@uax.es (S.H.M.); 2Department of Orthodontics, Faculty of Medicine and Dentistry, University of Salamanca, 37008 Salamanca, Spain; georgiatzironi@usal.es (G.T.); albertoalbaladejo@usal.es (A.A.M.)

**Keywords:** orthodontics, micro-screws, orthodontic anchorage, mini-implants, temporary anchorage devices

## Abstract

To analyze the influence of the computer-aided static navigation technique on the accuracy of placement of orthodontic micro-screws. One hundred and thirty-eight orthodontic micro-screws were randomly assigned to the following study groups: Group A. orthodontic micro-screw placement using a computer-aided static navigation technique (*n* = 69); B. orthodontic micro-screw placement using the conventional freehand technique (*n* = 69). In addition, the accuracy in the canine–premolar, premolar and molar sectors was analyzed in each study group. Cone-beam computed tomography and intraoral scans were taken both prior and subsequent to orthodontic micro-screw placement. The images were then uploaded using a 3D implant planning software, where the deviation and horizontal angles were analyzed using a multivariate linear model. These measurements were taken at the coronal entry point and apical endpoint between the planned orthodontic micro-screws. In addition, any complications resulting from micro-screw placement, such as spot perforations, were also analyzed in all dental sectors. The statistical analysis showed significant differences between the two study groups with regard to the coronal entry-point, apical end-point (*p* < 0.001) and angular deviations (*p* < 0.001) between the computer-aided static navigation technique and freehand technique study groups. Moreover, statistically significant differences were showed between the different dental sectors (*p* < 0.001). Additionally, twelve root perforations were observed at the conventional free hand technique study group while there were no root perforations in the computer-aided static navigation technique study group. The results showed that the computer-aided static navigation technique enables a more accurate orthodontic micro-screw placement with less intraoperative complications when compared with the conventional freehand technique.

## 1. Introduction

Anchorage systems pose a consistent issue in orthodontic treatments, as they are often uncomfortable, unattractive and their success relies heavily on patient cooperation [1]. The introduction of temporary anchorage devices (TAD) has drastically changed clinical treatment as they facilitate orthodontic treatments offering an alternative to conventional orthodontic treatments [2]. Currently, there are several anchored devices available for orthodontic purposes, the orthodontic micro-screws being the most popular due their small size characterized with smooth surfaces which allow the orthodontic micro-screws to be loaded immediately after their insertion as well as causing less post-operative pain and easy removal after treatment [3]. The anchorage can be classified according to the location being intra-oral, extra-oral or muscular; additionally, the anchorage can be also classified as simple, stationary or reciprocal, according to the applied force and even in single, compound, multiple and demands or minimum, moderate, maximum and absolute depending on the anchorage units [4]. Furthermore, temporary skeletal anchorage devices have been successfully used to provide intraoral absolute anchorage [5]. However, the success rate and intra-operative complications related to orthodontic micro-screws can be affected upon a number of variables, including the inherent characteristics attributed to the patient (age, gender, systematic diseases, periodontal status, smoking, skeletal pattern) [4], experience of the clinician [5], mechanical properties of the orthodontic micro-screw [6], patient care [7], placement torque [8], placement site [9,10], cortical bone thickness [8,9], insertion angle [11], root proximity [12], bone density [13], bone stress [14] and orthodontic force [12,13]. Moreover, root contact is considered one of the main drawbacks related to orthodontic micro-screw placement that it is possible to occur during insertion [7,10,12]. Therefore, some approaches have been proposed based on a cone-beam computed tomography (CBCT) scan [12], standard two-dimensional radiography [15] and panoramic radiography [16] to pre-operatively plan the insertion site of orthodontic micro-screws preventing root contact. Various insertion sites have been suggested according to the bone quality and low risk of root contact, such as edentulous areas, the palate and the zygomatic crest [17]; however, in most cases, the orthodontic micro-screws are inserted between the roots of contiguous teeth [17,18]. Unfortunately, complications derived from the orthodontic micro-screws are related to incorrect insertion positioning which may lead to the trauma of the periodontal ligament [7,12,18], artery or nerve injury and even maxillary sinus perforation [19]. In addition, potential root damage by orthodontic micro-screw placement has been linked to severe side-effects such as ankylosis, osteosclerosis and the loss of tooth vitality [7,10,12,18,20]. Therefore, it is mandatory to conduct an accurate pre-operative planning of the orthodontic micro-screw placement site before the insertion procedure [21]. Consequently, a custom-designed 3D-printed splint can be fabricated to facilitate a fully guided placement of orthodontic micro-screws [22].

The aim of this study was to analyze and evaluate the accuracy of orthodontic micro-screws and root contact prevalence, comparing a conventional freehand technique and a computer-aided static navigation technique in all dental sectors. The null hypothesis (H_0_) states that there would be no difference between the accuracy of orthodontic micro-screws between the conventional freehand technique and computer-aided static navigation technique at the coronal entry-point, apical end-point and angular deviation in all dental sectors.

## 2. Materials and Methods

### 2.1. Study Design

For the purposes of this study, 224 upper teeth from all dental sectors, extracted for periodontal and orthodontic reasons, were selected from cases treated at the Dental Centre of Innovation and Advanced Specialties at Alfonso X El Sabio University (Madrid, Spain) between February and April 2021. A randomized controlled in vitro study was carried out in compliance with the principles outlined by the German Ethics Committee’s statement on using organic tissues for medical research (Zentrale Ethikkommission, 2003). The study was authorized in November 2020 by the Ethical Committee of the Faculty of Health Sciences, University Alfonso X el Sabio (Madrid, Spain), in July 2021 (Process No. 21/2021). All patients gave the informed consent for their teeth to be used in the study.

### 2.2. Experimental Procedure

The teeth were embedded into fourteen experimental models of epoxy resin (Ref. 20-8130-128, EpoxiCure^®^, Buehler, IL, USA) with 16 teeth each. A silicone splint was created by a conventional impression to a dental training model of acrylic resin, and the teeth were placed onto it. Subsequently, the epoxy resin (Ref. 20-8130-128, EpoxiCure^®^, Buehler, IL, USA) was mixed following the manufacturer’s recommendations and poured inside the silicone splint with the teeth. After the epoxy resin set, the silicone splint was removed from the epoxy resin model. Assuming data distributed normally, to achieve a power of 80.00% to detect differences in the contrast of the null hypothesis H₀: μ₁ = μ₂ by means of a bilateral Student’s *t*-test for two independent samples, taking into account that the significance level as *p* < 0.05, it was necessary to include 138 orthodontic micro-screws. The orthodontic micro-screws (Dual Top^®^ Anchor System, JEIL Medical Corporation, Guro-gu, Seoul, Korea) were randomly divided (Epidat 4.1, Galicia, Spain) into the following study groups: Group A. orthodontic micro-screw placement in the incisive–canine sector by a computer-aided static navigation technique (NemoScan^®^, NEMOTEC, Madrid, Spain) (NAV-i) (*n* = 23), Group B. orthodontic micro-screw placement in the incisive–canine sector by conventional freehand technique (FHT-i) (*n* = 23), Group C. orthodontic micro-screw placement in the premolar sector by a computer-aided static navigation technique (NemoScan^®^, NEMOTEC, Madrid, Spain) (NAV-p) (*n* = 23), Group D. orthodontic micro-screw placement in the premolar sector by conventional freehand technique (FHT-p) (*n* = 23), Group E. Orthodontic micro-screw placement in the molar sector by a computer-aided static navigation technique (NemoScan^®^, NEMOTEC, Madrid, Spain) (NAV-m) (*n* = 23) and Group F. orthodontic micro-screw placement in the molar sector by conventional freehand technique (FHT-m) (*n* = 23). The teeth assigned to both experimental models presented similar anatomical dimensions evaluated with an electronic caliper and were positioned in the experimental model using a silicone splint to prevent different interradicular spaces between the different teeth of the experimental models.

A preoperative cone-beam computed tomography (CBCT) scan (WhiteFox, Acteón Médico-Dental Ibérica S.A.U., Satelec, Merignac, France) was taken of the experimental models of epoxy resin (Ref. 20-8130-128, EpoxiCure^®^, Buehler, IL, USA) using the following exposure parameters: 105.0 kilovolt peak, 8.0 milliamperes, 7.20 s, and a field of view of 15 × 13 mm (Figure 1A,B). A 3D surface scan was, subsequently, performed via 3D intraoral scan (True Definition, 3M ESPE™, Saint Paul, MN, USA) using 3D in-motion video imaging technology (Figure 1C). The datasets obtained from the digital workflow were added to 3D implant planning software (NemoScan^®^, NEMOTEC, Madrid, Spain) in order to plan the virtual placement of the orthodontic micro-screws (Ref. 16-G2-008, Dual Top^®^ Anchor System, JEIL Medical Corporation, Guro-gu, Seoul, Korea). The screws were 1.3 mm in diameter, 8.0 mm in length in the active part and 2.0 mm in the inactive part. Virtual placement was planned by matching the three-dimensional surface scan with CBCT data, with the key points being overlaid on the crown of the teeth (Figure 1D). Virtual orthodontic micro-screws were placed to a depth of 6 mm, an insertion angle of 90° to the longitudinal axis of the teeth, and a depth of 6.0 mm with respect to the cortical plate (Figure 1E).

The orthodontic micro-screw placement of the experimental model randomly sorted into the NAV study group were virtually planned on the 3D implant planning software (NemoScan^®^, NEMOTEC, Madrid, Spain). Afterwards, the surgical template was designed (Figure 1F) and manufactured (NemoScan^®^, NEMOTEC, Madrid, Spain) by 3D-printed techniques (Figure 1G). The interradicular spaces where the orthodontic micro-screws were placed were also randomly selected (Epidat 4.1, Galicia, Spain).

All experimental models were placed as the upper maxilla in a manikin by a specialized operator with a wide formation in self-tapping mini-screws and the orthodontic micro-screws (Dual Top^®^ Anchor System, JEIL Medical Corporation, Guro-gu, Seoul, Korea) randomly assigned to the FHT study group were placed in the experimental models by a unique operator per group with access to CBCT scan and the preoperative planning. According to the recommendations performed by Cozzani et al. [23] to place self-tapping orthodontic micro-screws after using a osteotomy pilot drill (Ref.: 112-MC.201, Dual Top^®^ Anchor System, JEIL Medical Corporation, Guro-gu, Seoul, Korea), he describes that the ideal insertion angle for self-tapping orthodontic micro-screws is 90° to provide the lowest stress values to the surrounding cancellous bone. All orthodontic micro-screws (Dual Top^®^ Anchor System, JEIL Medical Corporation, Guro-gu, Seoul, Korea) of both NAV and FHT study groups were inserted in the middle of the inter-root space at a distance of 2 mm from the alveolar ridge.

### 2.3. Measurement Procedure

After placing the orthodontic micro-screws (Dual Top^®^ Anchor System, JEIL Medical Corporation, Guro-gu, Seoul, Korea), postoperative CBCT scans of the experimental models were taken. Virtual orthodontic micro-screw (Dual Top^®^ Anchor System, JEIL Medical Corporation, Guro-gu, Seoul, Korea) planning and postoperative CBCT scans for the two study groups were added to the 3D implant planning software (NemoScan^®^, NEMOTEC, Madrid, Spain). These images were then matched to analyze the deviation angle (measured in the middle of the cylinder) and horizontal deviation (measured at the coronal entry-point and apical end-point) (Figure 2A–D) by an independent observer.

Root perforations after the orthodontic micro-screw (Dual Top^®^ Anchor System, JEIL Medical Corporation, Guro-gu, Seoul, Korea) placement were also analyzed and recorded in the 3D implant planning software (NemoScan^®^, NEMOTEC, Madrid, Spain) between the conventional freehand technique and computer-aided static navigation technique (Figure 3A–C).

### 2.4. Statistical Tests

All studied variables were recorded using SPSS 22.00 for Windows for statistical analysis. The descriptive statistical analysis was expressed as the mean and standard deviation (SD) of quantitative variables. A multivariate (generalized linear model (GLM)) was used for analyzing the effect of the study group, the dental group and the interaction between both variables in each of the response variables. In case of obtaining a significant result, post hoc pairwise comparisons were computed. To correct the type I error, the *p*-values were corrected using the Tukey correction. As the variables had normal distribution, *p* < 0.05 was determined statistically significant.

## 3. Results

The means and SD values for the coronal entry-point, apical end-point and angular deviation of the computer-aided static navigation technique and conventional freehand technique orthodontic micro-screws in all dental sectors are displayed in Table 1.

Statistically significant differences were shown between the computer-aided static navigation technique and conventional freehand technique study groups at the coronal entry point deviations of planned and performed orthodontic micro-screws (*p* < 0.001) (Figure 4).

In addition, the means and SD values for coronal entry-point deviations of the computer-aided static navigation technique and conventional freehand technique orthodontic micro-screws in the incisive–canine, premolar and molar dental sector are displayed in Table 2.

Statistically significant differences were also shown between the coronal entry-point deviations of the orthodontic micro-screws placed using the computer-aided static navigation technique and conventional freehand technique in the incisive–canine, premolar and molar dental sectors (*p* < 0.001) (Figure 5).

Specifically, the means and SD values for coronal entry-point deviations of the orthodontic micro-screws in the selected tooth positioning are displayed in Table 3 and Figure 6.

Statistically significant differences were also shown between the coronal entry-point deviations of the orthodontic micro-screws placed in the incisive–canine and premolar dental sectors (*p* = 0.001), incisive–canine and molar dental sectors (*p* < 0.001) and premolar and molar dental sectors (*p* < 0.001). The differences of the incisive–canine dental sector (F-value = 126.11) was higher than the premolar (F-value = 59.96) and molar dental sectors (F-value = 43.43) (Figure 6).

Additionally, statistically significant differences at the apical end-point deviations of planned and performed orthodontic micro-screws between the computer-aided static navigation technique and conventional freehand technique study groups are shown (*p* < 0.001) (Figure 7).

In addition, the means and SD values for the apical end-point deviation of the computer-aided static navigation technique and conventional freehand technique orthodontic micro-screws in the incisive–canine, premolar and molar dental sector are displayed in Table 4.

Statistically significant differences were also shown between the apical end-point deviations of the orthodontic micro-screws placed using the computer-aided static navigation technique and conventional freehand technique in the incisive–canine, premolar and molar dental sectors (*p* < 0.001) (Figure 8).

Specifically, the means and SD values for apical end-point deviations of the orthodontic micro-screws in the selected tooth positioning are displayed in Table 5 and Figure 9.

Statistically significant differences were also shown between the apical end-point deviations of the orthodontic micro-screws placed in the incisive–canine and premolar dental sectors (*p* = 0.001), incisive–canine and molar dental sectors (*p* < 0.001) and premolar and molar dental sectors (*p* < 0.001).

Furthermore, statistically significant differences in the angular deviations of planned and performed orthodontic micro-screws between the computer-aided static navigation technique and conventional freehand technique study groups are shown (*p* < 0.001) (Figure 10).

In addition, the means and SD values for the angular deviation of the computer-aided static navigation technique and conventional freehand technique orthodontic micro-screws in the incisive–canine, premolar and molar dental sector are displayed in Table 6.

Statistically significant differences were also shown between the angular deviations of the orthodontic micro-screws placed using the computer-aided static navigation technique and conventional freehand technique in the incisive–canine, premolar and molar dental sectors (*p* < 0.001) (Figure 11).

Specifically, the means and SD values for angular deviations of the orthodontic micro-screws in the selected tooth positioning are displayed in Table 7 and Figure 12.

Statistically significant differences were also shown between the angular deviations of the orthodontic micro-screws placed in the incisive–canine and premolar dental sectors (*p* = 0.001), incisive–canine and molar dental sectors (*p* < 0.001) and premolar and molar dental sectors (*p* < 0.001).

Twelve root perforations were observed in the conventional freehand technique study group after the orthodontic micro-screws placement at teeth 1.2, 1.4, 1.6, 1.7, 1.8, 2.4, 2.5, 2.6, 2.7 and 2.8, which matched with the highest coronal entry-point and apical end-point deviation values. No root perforations were observed in the computer-aided static navigation technique study group.

## 4. Discussion

The results of the present study rejected the null hypothesis (H_0_) which stated that there was no difference between the conventional freehand technique and computer-aided navigation technique at the coronal entry-point, apical end-point and angular deviation, nor in the intraoperative complications.

The present study showed higher deviations for the conventional freehand technique than the computer-aided static navigation technique at the coronal entry point, apical end-point and angular values. Previous studies have analyzed the importance of surgical templates in the accuracy of orthodontic micro-screw placement [22,24,25,26,27]; Cassetta et al., also showed similar results and reported that the surgical template considerably reduced the coronal, apical and angular deviations for the palatal micro-screw placement [28]. Moreover, Qiu et al., reported that the surgical templates used for orthodontic micro-screw placement provide a safer and more stable micro-screw insertion than the conventional freehand technique [29]. Even Suzuki reported promising results related to the accuracy of orthodontic micro-screws placed by the surgical template, although the results were analyzed using 2D periapical radiographs [22]. Some insertion sites of orthodontic micro-screws have been recommended to prevent the damage of root processes such as the zygomatic crest and mandibular buccal shelf area, although the most commonly used insertion sites are at the alveolar processes between dental roots [21]. Moreover, the orthodontic micro-screws can usually be inserted from the buccal side and it is commonly placed between the second premolar and first molar for maximum anchorage [20]. The interdental space between the second premolar and first molar at 5 mm from the alveolar crest is usually about 3.0 mm [20]. This space might be insufficient for an orthodontic micro-screw with a diameter ranging from 1.2 mm to 2.0 mm. Even though root contact can be prevented by the careful monitoring of the surgical procedure and the use of a radiograph, CT, or surgical stent, the orthodontic micro-screw might be close enough to the root to histologically affect the root surface and surrounding tissues [22].

Orthodontic micro-screws have reported a mean failure rate of 13.5%, which is a modestly small rate demonstrating their effectiveness in clinical practice [20]. Furthermore, one of the most common complication reported during orthodontic micro-screw insertion is causing a trauma to the dental root and/or the periodontal ligament; specifically, when the trauma is limited to the outer dental root surface without pulp involvement, it is less probably to influence the prognosis of the tooth [30]; in addition, the periodontal ligament and the cementum showed a complete reparation capacity between 12 and 18 weeks after the orthodontic micro-screw removal [31]. Moreover, when the orthodontic micro-screw insertion comprises the periodontal ligament, the patient begins to experience an increased sensation under local anesthesia [15,32]. Furthermore, if root contact occurs, the orthodontic micro-screws may require a greater insertion strength [31]. Finally, if the clinician suspects trauma to the tooth or periodontium, it is mandatory to immediately unscrew the orthodontic micro-screw two to three turns and assess the position radiographically [33]. In the present study, twelve orthodontic micro-screws placed by conventional freehand technique caused root perforation and none root contact was shown in the teeth randomly assigned to the computer-aided static navigation technique. In addition, Kalra et al. analyzed the planning performed by the CBCT scan and 2D radiograph to prevent root perforations, and concluded that the planning performed by the CBCT scan showed no root perforations and the planning performed by the 2D radiograph showed three root perforations in twenty patients [21]. Moreover, Bufalá Perez et al. analyzed the influence of clinician experience on the accuracy of the placement of orthodontic self-tapping micro-screws and reported five out of thirty root perforations in the study group with no experience, compared with no root perforation in the group placed by an orthodontist with 10 years’ experience [5].

The orthodontic micro-screw placement between contiguous roots necessitates a proper radiographic planning, including a surgical template, as well as panoramic and periapical radiographs in order to determinate the safest site of placement [16,34,35,36,37,38,39].

Severe bone damage during the insertion of orthodontic micro-screws can result in bone remodeling and induce root resorption. If the periodontal ligature is severely injured and bone grows toward the reabsorbed root, the ligature is not able to protect the root and may lead to tooth ankylosis [40]. In addition, root resorption can be triggered by stimulating the activation of the periodontal ligament in differentiating cementoclasts. The incidence of root resorption can be limited if minimal injury is experienced during the orthodontic micro-screw insertion procedure. A seemingly heavy injury insertion may induce root resorption even though there is no proximity of the orthodontic micro-screws and the root [31]. Motoyoshi et al. categorizes the root proximity of orthodontic micro-screws into three groups: A. no contact between the root and orthodontic micro-screw, B. one point of contact between the root and the orthodontic micro-screws and C. two or more points of contact [41]. Moreover, in the most severe cases, causing the loss of pulp vitality, ankylosis and root resorption are rare complications. Finally, the risk of pathology increases rapidly when orthodontic micro-screws are more proximal to the dental root surface, with a critical proximity found to be 1 mm. For those reasons, it is important to predict the accurate position of the orthodontic micro-screws, because as well as tissue damage, the contact of orthodontic micro-screws to the root may also provoke the loss of orthodontic micro-screw stability [42].

The present study had the strength of including a sample size higher than the previous studies of Qiu et al. (*n* = 30) [25], Liu et al. (*n* = 34) [24], Miyazawa et al. (*n* = 44) [27] and Bae et al. (*n* = 45) [26], as well as presenting the results regarding the dental sector where the micro-screws were placed. This methodology aimed to establish more reliable results as far as it concerns the morphology of the roots in those specific areas and the interdental distance which differs between each dental sector. On the other hand, it was an in vitro study with extracted teeth.

## 5. Conclusions

In conclusion, bearing in mind the limitations of this in vitro study, the results showed that the computer-aided static navigation technique had an effect on the accuracy of the orthodontic micro-screw placement, resulting in fewer intraoperative complications.

## Figures and Tables

**Figure 1 jcm-10-04127-f001:**
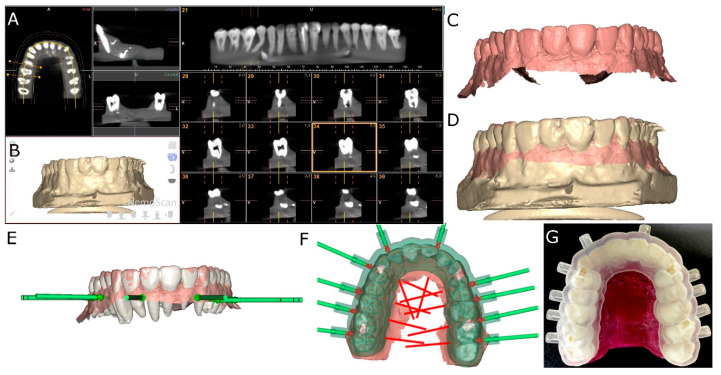
(**A**,**B**) DICOM files from the CBCT scan, (**B**) CBCT scan rendering, (**C**) STL digital file from the digital impression, (**D**) alignment procedure between STL and CBCT scan digital files, (**E**) orthodontic micro-screws planning position, (**F**), surgical template design and (**G**) manufacturing.

**Figure 2 jcm-10-04127-f002:**
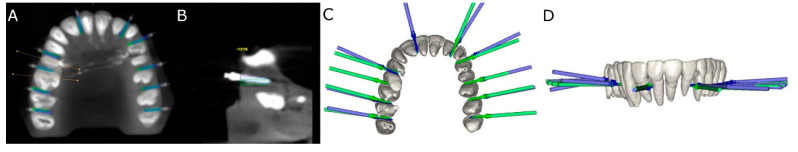
(**A**–**D**) Deviations measurement procedure between planned (green cylinder) and performed (blue cylinder) orthodontic micro-screws in the computer-aided static navigation technique study group.

**Figure 3 jcm-10-04127-f003:**
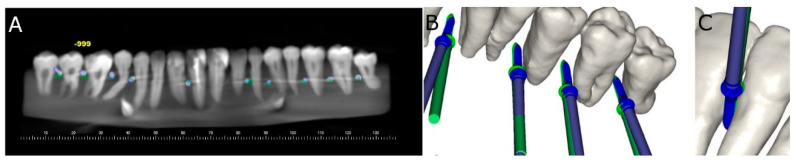
(**A**) Radiographic analysis of the root perforation in the 3D implant planning software, (**B**) relationship between the root processes and the planned (green micro-screw) and performed (blue micro-screw) orthodontic micro-screws using the computer-aided static navigation technique and (**C**) by the conventional freehand technique.

**Figure 4 jcm-10-04127-f004:**
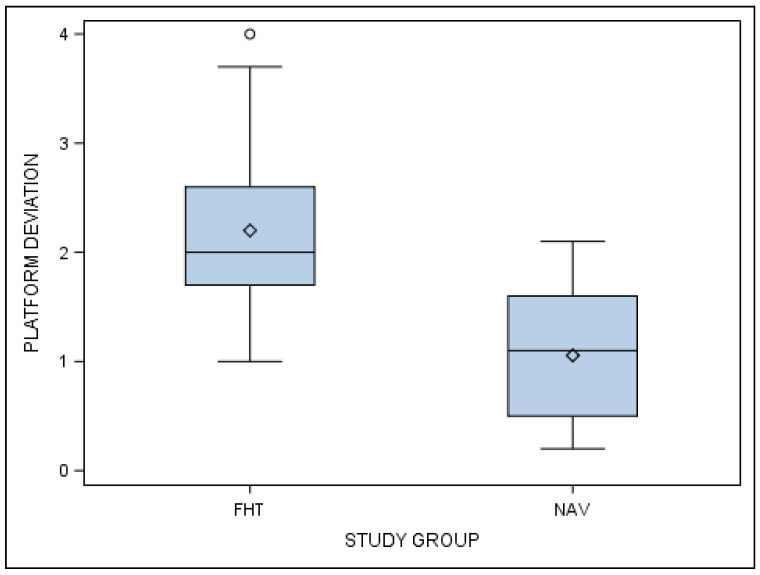
Box plot of the coronal deviations in planned and performed orthodontic micro-screws between computer-aided static navigation technique and conventional freehand technique study groups.

**Figure 5 jcm-10-04127-f005:**
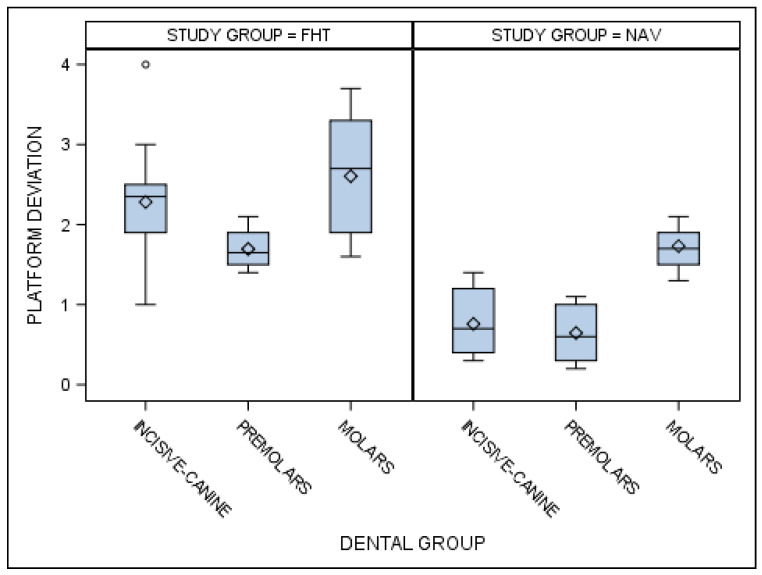
Box plot of the coronal entry-point deviations of the orthodontic micro-screws placed using the computer-aided static navigation technique and conventional freehand technique in the incisive–canine, premolar and molar dental sectors.

**Figure 6 jcm-10-04127-f006:**
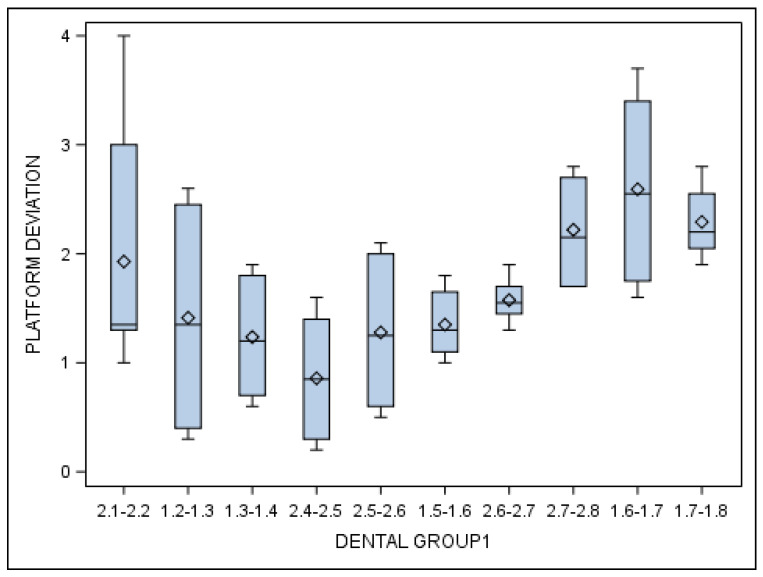
Box plot of the coronal entry-point deviations of the orthodontic micro-screws in the selected tooth positioning.

**Figure 7 jcm-10-04127-f007:**
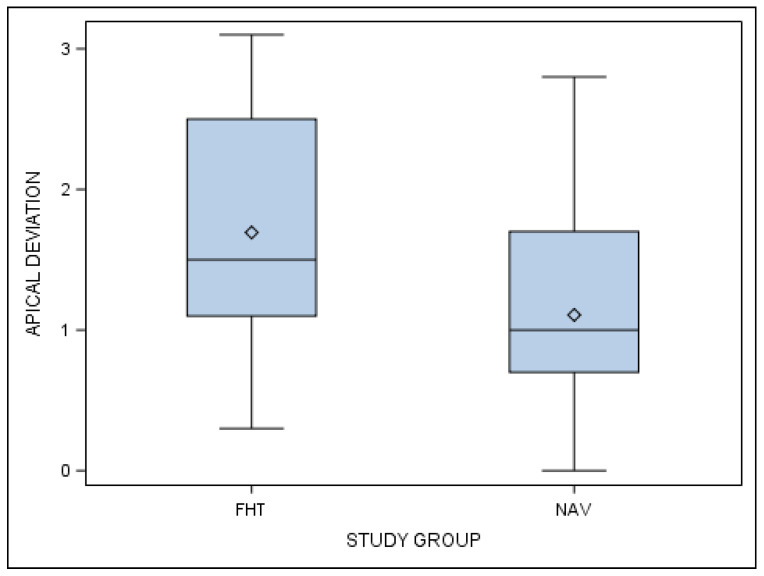
Box plot of the apical deviations in planned and performed orthodontic micro-screws between computer-aided static navigation technique and conventional freehand technique study groups.

**Figure 8 jcm-10-04127-f008:**
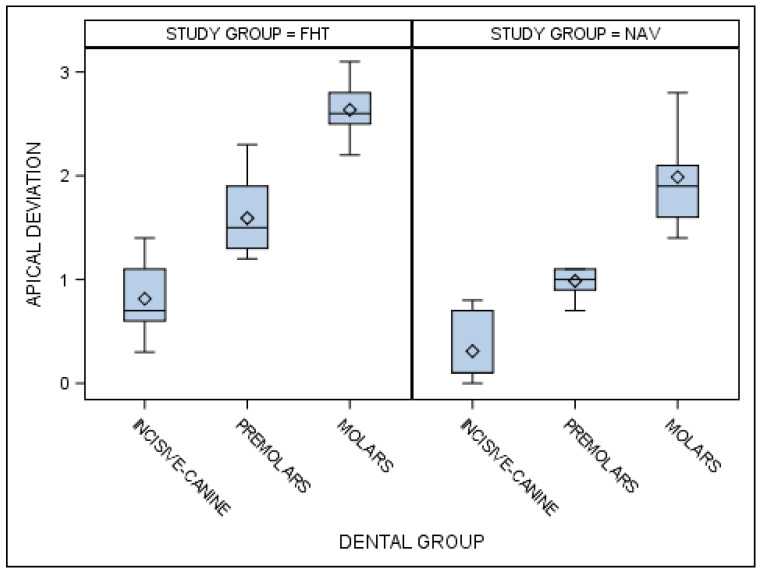
Box plot of the apical end-point deviations of the orthodontic micro-screws placed using the computer-aided static navigation technique and conventional freehand technique in the incisive–canine, premolar and molar dental sectors.

**Figure 9 jcm-10-04127-f009:**
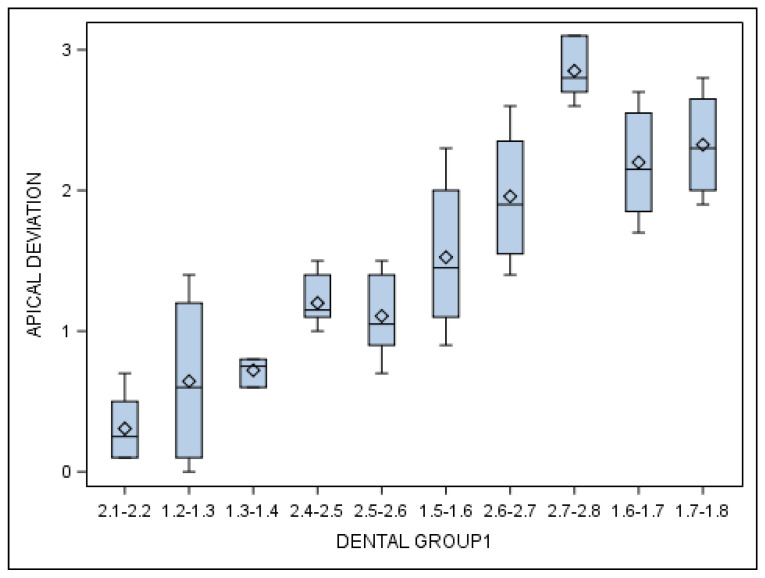
Box plot of the apical end-point deviations of the orthodontic micro-screws in the selected tooth positioning.

**Figure 10 jcm-10-04127-f010:**
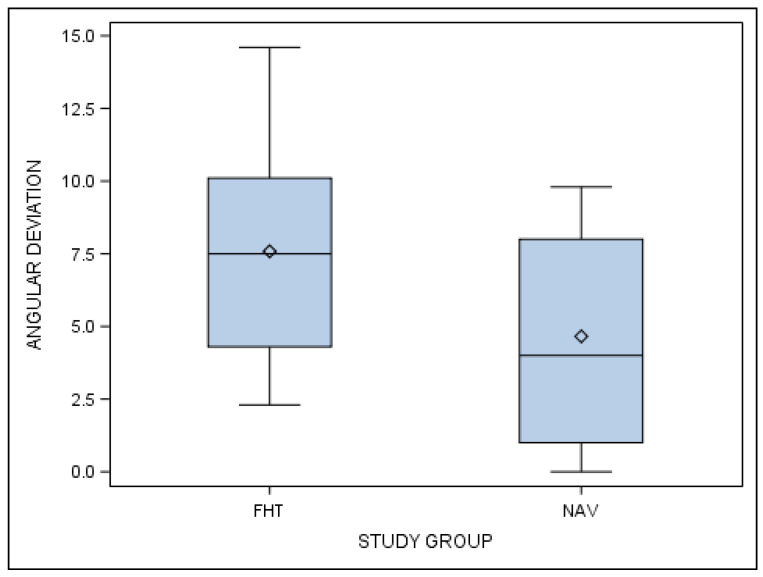
Box plot of angular deviations in planned and performed orthodontic micro-screws between the computer-aided static navigation technique and conventional freehand technique study groups.

**Figure 11 jcm-10-04127-f011:**
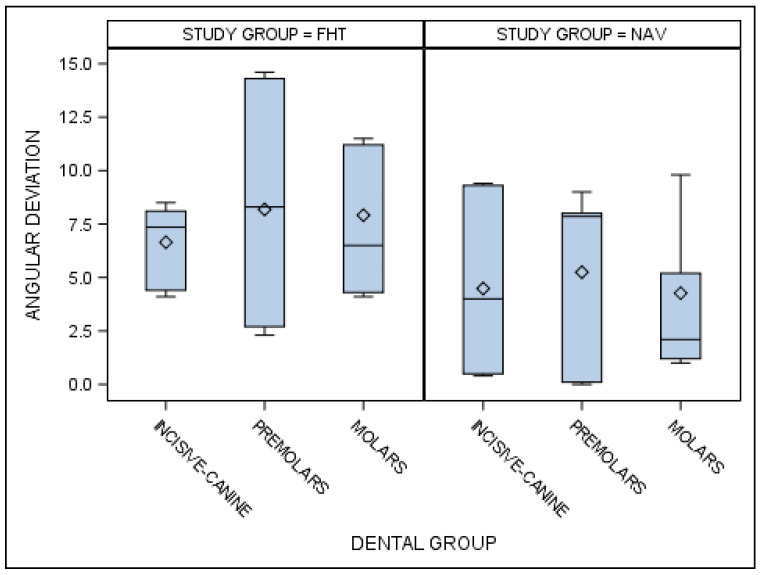
Box plot of the angular deviations of the orthodontic micro-screws placed using the computer-aided static navigation technique and conventional freehand technique in the incisive–canine, premolar and molar dental sectors.

**Figure 12 jcm-10-04127-f012:**
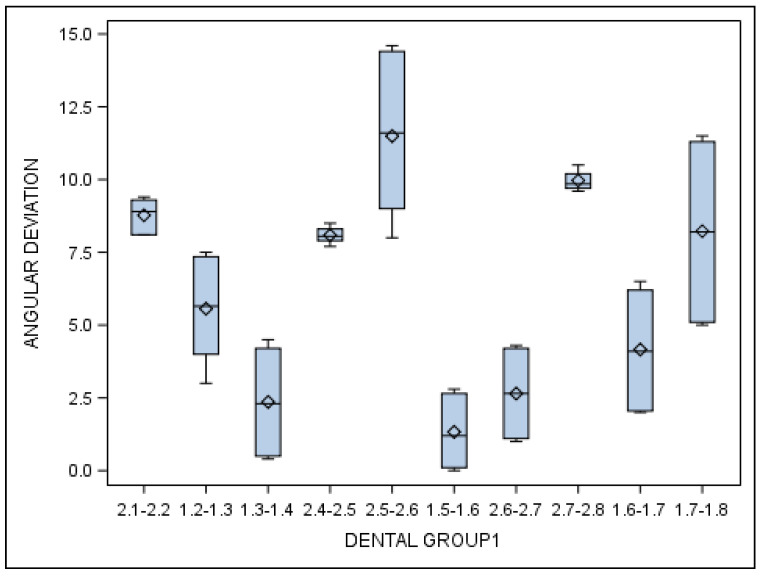
Box plot of the angular deviations of the orthodontic micro-screws in the selected tooth positioning.

**Table 1 jcm-10-04127-t001:** Descriptive deviation values at coronal entry-point (mm), apical end-point (mm), and angular (°) levels of the orthodontic micro-screws placed by using conventional freehand technique and computer-aided static navigation technique study groups.

Location	Study Group	*n*	Mean	SD	Minimum	Maximum
Coronal	NAV	69	1.06	0.59	0.20	2.10
	FHT	69	2.20	2.00	1.00	4.00
Apical	NAV	69	1.11	0.77	0.10	2.80
	FHT	69	1.69	0.82	0.40	3.10
Angular	NAV	69	4.66	3.65	0.00	9.80
	FHT	69	7.58	3.50	2.30	14.60

**Table 2 jcm-10-04127-t002:** Descriptive deviation values at coronal entry-point (mm) of the orthodontic micro-screws placed by using conventional freehand technique and computer-aided static navigation technique study groups in the incisive–canine, premolar and molar dental sector.

Tooth Location	Study Group	*n*	Mean	SD	Minimum	Maximum
Incisive–canine	NAV	23	0.76	0.39	0.30	1.40
	FHT	23	2.28	0.63	1.00	4.00
Premolar	NAV	23	0.65	0.35	0.20	1.10
	FHT	23	1.70	0.25	1.40	2.10
Molar	NAV	23	1.73	0.24	1.30	2.10
	FHT	23	2.60	0.65	1.60	3.70

**Table 3 jcm-10-04127-t003:** Descriptive deviation values at coronal entry-point (mm) of the orthodontic micro-screws placed in the selected tooth positioning.

Tooth Location	*n*	Mean	SD	Minimum	Maximum
2.1-2.2	14	1.93	0.94	1.00	4.00
1.2-1.3	14	1.41	1.07	0.30	2.60
1.3-1.4	13	1.24	0.59	0.60	1.90
2.4-2.5	14	0.86	0.63	0.20	1.60
2.5-2.6	14	1.28	0.75	0.50	2.10
1.5-1.6	14	1.35	0.31	1.00	1.80
2.6-2.7	13	1.58	0.19	1.30	1.90
2.7-2.8	14	2.22	0.51	1.70	2.80
1.6-1.7	14	2.59	0.90	1.60	3.70
1.7-1.8	14	2.29	0.30	1.90	2.80

**Table 4 jcm-10-04127-t004:** Descriptive deviation values at apical end-point (mm) of the orthodontic micro-screws placed by using conventional freehand technique and computer-aided static navigation technique study groups in the incisive–canine, premolar and molar dental sector.

Tooth Location	Study Group	*n*	Mean	SD	Minimum	Maximum
Incisive–canine	NAV	23	0.31	0.32	0.00	0.80
	FHT	23	0.81	0.34	0.30	1.40
Premolar	NAV	23	0.99	0.12	0.70	1.10
	FHT	23	1.59	0.35	1.20	2.30
Molar	NAV	23	1.99	0.43	1.40	2.80
	FHT	23	2.63	0.25	2.20	3.10

**Table 5 jcm-10-04127-t005:** Descriptive deviation values at apical end-point (mm) of the orthodontic micro-screws placed in the selected tooth positioning.

Tooth Location	*n*	Mean	SD	Minimum	Maximum
2.1-2.2	14	0.31	0.21	0.10	0.70
1.2-1.3	14	0.64	0.59	0.00	1.40
1.3-1.4	13	0.72	0.09	0.60	0.80
2.4-2.5	14	1.20	0.17	1.00	1.50
2.5-2.6	14	1.11	0.29	0.70	1.50
1.5-1.6	14	1.53	0.51	0.90	2.30
2.6-2.7	13	1.96	0.46	1.40	2.60
2.7-2.8	14	2.85	0.20	2.60	3.10
1.6-1.7	14	2.20	0.40	1.70	2.70
1.7-1.8	14	2.33	0.35	1.90	2.80

**Table 6 jcm-10-04127-t006:** Descriptive deviation values at angular level (°) of the orthodontic micro-screws placed by using conventional freehand technique and computer-aided static navigation technique study groups in the incisive–canine, premolar and molar dental sector.

Tooth Location	Study Group	*n*	Mean	SD	Minimum	Maximum
Incisive–canine	NAV	23	4.48	3.67	0.40	9.40
	FHT	23	6.65	1.72	4.10	8.50
Premolar	NAV	23	5.25	4.03	0.00	9.00
	FHT	23	8.18	4.98	2.30	14.60
Molar	NAV	23	4.27	3.32	1.00	9.80
	FHT	23	7.91	2.99	4.10	11.50

**Table 7 jcm-10-04127-t007:** Descriptive deviation values at angular level (°) of the orthodontic micro-screws placed in the selected tooth positioning.

Tooth Location	*n*	Mean	SD	Minimum	Maximum
2.1-2.2	14	8.77	0.59	8.10	9.40
1.2-1.3	14	5.56	1.90	3.00	7.50
1.3-1.4	13	2.36	1.96	0.40	4.50
2.4-2.5	14	8.09	0.27	7.70	8.50
2.5-2.6	14	11.49	3.05	8.00	14.60
1.5-1.6	14	1.33	1.32	0.00	2.80
2.6-2.7	13	2.65	1.62	1.00	4.30
2.7-2.8	14	9.97	0.31	9.60	10.50
1.6-1.7	14	4.16	2.20	2.00	6.50
1.7-1.8	14	8.23	3.25	5.00	11.50

## Data Availability

Data available on request owing to restrictions, e.g., privacy or ethical.

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
