# Peer review of "Influence of the Computer-Aided Static Navigation Technique on the Accuracy of the Orthodontic Micro-Screws Placement: An In Vitro Study"

_jcm, 2021, doi:10.3390/jcm10184127_

Round 1
Reviewer 1 Report
In this manuscript, the authors investigated the effects of computer-aided static navigation technique on the accuracy of TADs placement using an in vitro model. Although it involves an interesting and clinically relevant topic, the present study has several concerns, primarily in terms of its novelty and methodology.
Major concerns:
- Multiple in vivo studies have been done about ten years ago to test the accuracy of TADs placement with the 3D surgical guide, and showed that the use of a surgical guide was associated with relatively precise placement and a low risk of root injuries. Examples include Liu, et al., 2010, AJODO; Suzuki, et al., 2008, Journal of oral and maxillofacial surgery; Miyazawa, 2010, EJO. These in vivo studies provided more direct and stronger evidence than in vitro Moreover, even in vitro studies similar to the present manuscript have also been published already, including Mi-JuBae et al, AJODO, 2013; Qiu et al., 2012, Angle Orthodontist. Therefore, this manuscript lacks adequate originality and novelty.
- The methodology raised a couple of questions:
- The authors stated that “The teeth were embedded into two experimental models of epoxy resin with 16 teeth each” and 20 TADs were randomly distributed into NAV and FHT groups with 10 in each group (Page 2). The figures showed 10 TADs in each model. Therefore, it seems that NAV TADs were placed in one resin model and FHT TADs were placed in the other resin model, although the authors didn’t clarify. If they did so, the interradicular space in these two models would be different, which would affect the likelihood of root injuries.
- Small sample size (10 TADs per group). Most of the above-mentioned studies had a larger sample size.
- According to the figures, 10 interradicular TADs were placed not only among posterior teeth, but also among anterior teeth. We know the interradicular spaces in posterior and anterior regions are different. The authors only stated FHT group had 4 root injuries and have higher coronal entry-point deviation, but didn’t specify at which interradicular spaces these injuries and deviations mostly happened.
- The authors didn’t report where the TADs were inserted (e.g., how far from the alveolar ridge) or the distance of the interradicular space at the insertion sites.
Author Response
Dear Reviewer,
I’m pleased to resubmit the manuscript of the work entitled, “Influence of the Computer-Aided Static Navigation Technique on the Accuracy of the Orthodontic Micro-screws Placement
Reviewer 1: English language and style are fine/minor spell check required
Response: In order to adapt to the reviewer's 1 comments, we have send the manuscript to a specialized English Editing Service.
Reviewer 1: Multiple in vivo studies have been done about ten years ago to test the accuracy of TADs placement with the 3D surgical guide, and showed that the use of a surgical guide was associated with relatively precise placement and a low risk of root injuries. Examples include Liu, et al., 2010, AJODO; Suzuki, et al., 2008, Journal of oral and maxillofacial surgery; Miyazawa, 2010, EJO. These in vivo studies provided more direct and stronger evidence than in vitro Moreover, even in vitro studies similar to the present manuscript have also been published already, including Mi-JuBae et al, AJODO, 2013; Qiu et al., 2012, Angle Orthodontist. Therefore, this manuscript lacks adequate originality and novelty.
Response: In order to adapt to the reviewer's 1 comments, we clarify that the study of Suzuki et al analyzed the accuracy of orthodontic micro-screws using 2D periapical radiographs but the present study used 3D images from the CBCT scan, which allows an accurate and objective three-dimensional analysis in all space planes.
Liu et al and Miyazawa et al, provided results of the accuracy of orthodontic micro-screws between the roots of 2nd premolar and 1st molar. However, the present study provides information of the accuracy of orthodontic micro-screws between all roots. In the same way, the study of Qui et al involves orthodontic micro-screws inserted in the maxillary or mandibular molar area.
In addition, the study of Mi-Ju et al used 2D periapical radiographs to determine the orthodontic micro-screws position assigned to the control group.
As a result, the present study has significant differences compared to the ones mentioned above, and adds further information to the data selection.
Reviewer 1: The methodology raised a couple of questions: The authors stated that “The teeth were embedded into two experimental models of epoxy resin with 16 teeth each” and 20 TADs were randomly distributed into NAV and FHT groups with 10 in each group (Page 2). The figures showed 10 TADs in each model. Therefore, it seems that NAV TADs were placed in one resin model and FHT TADs were placed in the other resin model, although the authors didn’t clarify. If they did so, the interradicular space in these two models would be different, which would affect the likelihood of root injuries.
Response: In order to adapt to the reviewer's 1 comments, we have clarified the question in the “Material and Method” section.
Reviewer 1: Small sample size (10 TADs per group). Most of the above-mentioned studies had a larger sample size.
Response: In order to adapt to the reviewer's 1 comments, we clarify that the sample size selected for this study is based on previous studies: Zubizarreta-Macho A, Riad-Deglow E, O´Connor Esteban M, Hernández Montero S, Tzironi G, Abella Sans F, Albaladejo Martínez A. Novel Digital Technique to Analyze the Accuracy and Intraoperative Complications of Orthodontic Self-tapping and Self-drilling Micro-screws Placement Techniques: An in Vitro Study. AJODO. 2021, Epub of Print and Bufalá Pérez, M.; O’Connor Esteban, M.; Zubizarreta-Macho, Á.; Riad Deglow, E.; Hernández Montero, S.; Abella Sans, F.; Albaladejo Martínez, A. Novel Digital Technique to Analyze the Influence of theOperator Experience on the Accuracy of the Orthodontic Micro-Screws Placement. Appl. Sci. 2021, 11, 400. In addition, the sample size showed a power of 88.4 which is considered acceptable.
Reviewer 1: According to the figures, 10 interradicular TADs were placed not only among posterior teeth, but also among anterior teeth. We know the interradicular spaces in posterior and anterior regions are different.
Response: In order to adapt to the reviewer's 1 comments, we clarify that the aim of the study was to analyze and compare the accuracy of orthodontic micro-screws between conventional freehand technique and computer-aided static navigation technique and the root contact prevalence of both orthodontic micro-screws placement techniques, regardless the region; moreover, the experimental models assigned to each study group were prepared with teeth with similar measurements placed in the same positions.
Reviewer 1: The authors only stated FHT group had 4 root injuries and have higher coronal entry-point deviation, but didn’t specify at which interradicular spaces these injuries and deviations mostly happened.
Response: In order to adapt to the reviewer's 1 comments, we have clarified the tooth affected by root perforation and also the deviation type.
Reviewer 1: The authors didn’t report where the TADs were inserted (e.g., how far from the alveolar ridge) or the distance of the interradicular space at the insertion sites.
Response: In order to adapt to the reviewer's 1 comments, we have clarified the insertion characteristics in the Material and Methods section.
We take this opportunity to thank the recommendations and suggestions made by the reviewers to improve the document.
Yours sincerely,
Authors
Reviewer 2 Report
Dear authors, thank you very much for your paper.
In this paper the authors presented an in vitro study entitled: “Influence of the Computer-Aided Static Navigation Technique on the Accuracy of the Orthodontic Micro-screws Placement” to determine and compare the accuracy of orthodontic micro-screws between conventional freehand technique and computer-aided static navigation technique and the root contact prevalence of both orthodontic micro-screws placement techniques;
It is strongly recommended that the whole text must be revised by a native English speaker. Mistakes and typos are present in the whole text (Es. Line 130). Please revise carefully it. Please specify the type of the study (in vitro) in the title and in the abstract section to be more understandable to the readers.
In the introduction section the issue is well addressed, but, despite this, in the section concerning “there are several anchored devices available for orthodontic purposes..” , the authors must increase the introduction section by adding studies in this regard.
Although the manuscript deals with an interesting issue, different issues of the manuscript must be revised and improved prior to be published in this journal.
In fact, the discussion sections seems quite poor. What can be seen in the literature about it? What studies have been carried out with this technique?
Please investigate in the literature by adding similar studies for the discussion section:
https://doi.org/10.1016/j.ijom.2018.03.018
10.3760/cma.j.issn.1002-0098.2016.06.004
Definitely, I think that an in vitro work on the subject of micro screws, although overall well written, is too minimal to be published in a journal like JCM (impact factor 4.241).
Author Response
Dear Reviewer,
I’m pleased to resubmit the manuscript of the work entitled, “Influence of the Computer-Aided Static Navigation Technique on the Accuracy of the Orthodontic Micro-screws Placement
Reviewer 2: English language and style are fine/minor spell check required
Response: In order to adapt to the reviewer's 2 comments, we have send the manuscript to a specialized English Editing Service.
Reviewer 2: It is strongly recommended that the whole text must be revised by a native English speaker.
Response: In order to adapt to the reviewer's 2 comments, we have send the manuscript to a specialized English Editing Service.
Reviewer 2: Mistakes and typos are present in the whole text (Es. Line 130).
Response: In order to adapt to the reviewer's 2 comments, we have revised the full manuscript.
Reviewer 2: Please specify the type of the study (in vitro) in the title and in the abstract section to be more understandable to the readers
Reviewer 2: In the introduction section the issue is well addressed, but, despite this, in the section concerning “there are several anchored devices available for orthodontic purposes..” , the authors must increase the introduction section by adding studies in this regard.
Response: In order to adapt to the reviewer's 2 comments, we have increased the Introduction section.
Reviewer 2: The discussion sections seems quite poor. What can be seen in the literature about it? What studies have been carried out with this technique?
Response: In order to adapt to the reviewer's 2 comments, we have improved the Discussion section by adding some related articles and discussed with the present results.
Reviewer 2: Please investigate in the literature by adding similar studies for the discussion section: https://doi.org/10.1016/j.ijom.2018.03.018, 10.3760/cma.j.issn.1002-0098.2016.06.004.
Response: In order to adapt to the reviewer's 2 comments, we have improved the Discussion section by adding some related articles.
We take this opportunity to thank the recommendations and suggestions made by the reviewers to improve the document.
Yours sincerely,
Authors
Reviewer 3 Report
I read with great interest the Manuscript titled “Influence of the Computer-Aided Static Navigation Technique on the Accuracy of the Orthodontic Micro-screws Placement” (ID: jcm-1290798), which falls within the aim of Journal of Clinical Medicine.
Authors should consider the following recommendations:
- How sample size was calculated? Please, specify in Materials and Methods section;
- How was confirmed the homogeneity of samples to avoid eventual differences between the control and experimental groups?
- Manufacturing of “experimental models of epoxy resin” should be described more in detail;
- Legend of Figure 1: “(B) 3D reconstruction if the CBCT scan”, please revise this sentence;
- The positioning of micro-screws should be explained more in detail (in correspondence of which teeth have been placed the micro-screws and why?);
- The discussion seems to be a general description of micro-screws. Instead, it should be mainly focused on the strengths and limitations of the methodology, comparison of the results obtained with other similar studies, clinical implications of the results obtained. Please, revise this section according to these indications.
Author Response
Dear Reviewer,
I’m pleased to resubmit the manuscript of the work entitled, “Influence of the Computer-Aided Static Navigation Technique on the Accuracy of the Orthodontic Micro-screws Placement
Reviewer 3: Moderate English changes required
Response: In order to adapt to the reviewer's 3 comments, we have send the manuscript to a specialized English Editing Service.
Reviewer 3: How sample size was calculated? Please, specify in Materials and Methods section;
Response: In order to adapt to the reviewer's 3 comments, we clarify that the sample size selected for this study is based on previous studies: Zubizarreta-Macho A, Riad-Deglow E, O´Connor Esteban M, Hernández Montero S, Tzironi G, Abella Sans F, Albaladejo Martínez A. Novel Digital Technique to Analyze the Accuracy and Intraoperative Complications of Orthodontic Self-tapping and Self-drilling Micro-screws Placement Techniques: An in Vitro Study. AJODO. 2021, Epub of Print and Bufalá Pérez, M.; O’Connor Esteban, M.; Zubizarreta-Macho, Á.; Riad Deglow, E.; Hernández Montero, S.; Abella Sans, F.; Albaladejo Martínez, A. Novel Digital Technique to Analyze the Influence of theOperator Experience on the Accuracy of the Orthodontic Micro-Screws Placement. Appl. Sci. 2021, 11, 400. In addition, the sample size showed a power of 88.4 which is considered acceptable.
Reviewer 3: How was confirmed the homogeneity of samples to avoid eventual differences between the control and experimental groups?
Response: In order to adapt to the reviewer's 3 comments, we have added the following sentence to the Material and Methods section: “The teeth assigned to both experimental models presented the same anatomy and were positioned in the experimental model using a silicone splint to prevent different interradicular spaces between the different teeth of the experimental models”.
Reviewer 3: Manufacturing of “experimental models of epoxy resin” should be described more in detail;
Response: In order to adapt to the reviewer's 3 comments, we have described in detail the manufacturing process of the models.
Reviewer 3: Legend of Figure 1: “(B) 3D reconstruction if the CBCT scan”, please revise this sentence;
Response: In order to adapt to the reviewer's 3 comments, we have removed the word “3D”.
Reviewer 3: The positioning of micro-screws should be explained more in detail (in correspondence of which teeth have been placed the micro-screws and why?);
Response: In order to adapt to the reviewer's 3 comments, we have described how the interradicular spaces were selected in the Material and Methods section: “The interradicular spaces where the orthodontic micro-screws were placed were also randomly selected (Epidat 4.1, Galicia, Spain)”.
Reviewer 3: The discussion seems to be a general description of micro-screws. Instead, it should be mainly focused on the strengths and limitations of the methodology, comparison of the results obtained with other similar studies, clinical implications of the results obtained. Please, revise this section according to these indications.
Response: In order to adapt to the reviewer's 3 comments, we have increased the Discussion section: “The present study showed higher deviations for the conventional freehand technique than the computer-aided static navigation technique at the coronal entry point, apical end-point and angular values. Previous studies have analyzed the importance of surgical templates in the accuracy of orthodontic micro-screws placement [25,29-32], Cassetta et al. also showed similar results and reported that the surgical template reduced considerably the coronal, apical and angular deviations for palatal micro-screws placement [27]. Moreover, Qiu et al. reported that the surgical templates used for orthodontic micro-screws placement provide safer and more stable micro-screws insertion than conventional freehand technique [28]. Even Suzuki reported promising results related to the accuracy of orthodontic micro-screws placed by surgical template, although the results were analyzed using 2D-periapical radiographs [25].
We take this opportunity to thank the recommendations and suggestions made by the reviewers to improve the document.
Yours sincerely,
Authors
Round 2
Reviewer 1 Report
Thank you for the revision.
The authors have improved the quality of presentation, including having clarified the location of miniscrew placement, the locations of injuries, etc.
However, the lack of novelty and small sample size are still the biggest drawbacks of this study. Most of the old studies I included in my first report did use CBCT scan and 3D surgical splint; their in vivo approaches provided stronger evidence; the samples sizes in these studies are much larger than the present study.
With such a small sample size, “the present study provides information of the accuracy of orthodontic micro-screws between all roots” doesn’t seem to be an advantage. Each group only has two miniscrews placed in the anterior region. The authors are commended to increase the samples size, and analyze the molar and anterior regions separately.
Therefore, although the quality of the presentation has improved, the novelty and quality of this study are too minimal to be published in JCM (IF=4.241). Authors are recommended to send this revised version to a dental journal.
Author Response
Dear Reviewer 1,
I’m pleased to resubmit the manuscript of the work entitled, “Influence of the Computer-Aided Static Navigation Technique on the Accuracy of the Orthodontic Micro-screws Placement. An in Vitro Study”.
Reviewer 1: English language and style are fine/minor spell check required
Response: In order to adapt to the reviewer's 1 comments, we send the manuscript to a specialized English Editing Service.
Reviewer 1: However, the lack of novelty and small sample size are still the biggest drawbacks of this study. Most of the old studies I included in my first report did use CBCT scan and 3D surgical splint; their in vivo approaches provided stronger evidence; the samples sizes in these studies are much larger than the present study. With such a small sample size, “the present study provides information of the accuracy of orthodontic micro-screws between all roots” doesn’t seem to be an advantage. Each group only has two miniscrews placed in the anterior region. The authors are commended to increase the samples size, and analyze the molar and anterior regions separately.
Response: In order to adapt to the reviewer's 1 comments, we clarify that we have newly assessed the sample size in base of the results of the previous study, to obtain statistically significant differences: to achieve a power of 80.00% to detect differences in the contrast of the null hypothesis H₀: μ₁ = μ₂ by means of a bilateral Student's T-test for two independent samples, taking into account that the significance level is 5.00% , and assuming that the mean of the Reference group is 1.67 mm, the mean of the Experimental group is 1.25 mm and the standard deviation of both groups is 0.87 mm, it will be necessary to include 69 experimental units in the Reference group and 69 units in the Experimental group, totalling 138 experimental units in the study; which is higher than Qiu et al study (n = 30), Liu et al (n = 34), Miyazawa et al (n = 44) and Bae et al (n = 45).
Additionally, the results were assessed regarding the dental sector that was placed (incisive-canine, premolar and molar) to provide novel and useful information that have not been provided in previous studies.
We take this opportunity to thank the recommendations and suggestions made by the reviewers to improve the document.
Yours sincerely,
Authors
Reviewer 2 Report
Dear Authors,
The authors have increased the scientific quality of the article by adding the required information and now the paper is well-written, however the study objective of this paper remains non-innovative and original.
Author Response
Dear Reviewer 2,
I’m pleased to resubmit the manuscript of the work entitled, “Influence of the Computer-Aided Static Navigation Technique on the Accuracy of the Orthodontic Micro-screws Placement. An in Vitro Study”.
Reviewer 2: English language and style are fine/minor spell check required
Response: In order to adapt to the reviewer's 2 comments, we send the manuscript to a specialized English Editing Service.
Reviewer 2: The authors have increased the scientific quality of the article by adding the required information and now the paper is well-written, however the study objective of this paper remains non-innovative and original.
Response: In order to adapt to the reviewer's 2 comments, we clarify that we have newly assessed the sample size in base of the results of the previous study, to obtain statistically significant differences: to achieve a power of 80.00% to detect differences in the contrast of the null hypothesis H₀: μ₁ = μ₂ by means of a bilateral Student's T-test for two independent samples, taking into account that the significance level is 5.00% , and assuming that the mean of the Reference group is 1.67 mm, the mean of the Experimental group is 1.25 mm and the standard deviation of both groups is 0.87 mm, it will be necessary to include 69 experimental units in the Reference group and 69 units in the Experimental group, totalling 138 experimental units in the study; which is higher than Qiu et al study (n = 30), Liu et al (n = 34), Miyazawa et al (n = 44) and Bae et al (n = 45).
Additionally, the results were assessed regarding the dental sector that was placed (incisive-canine, premolar and molar) to provide novel and useful information that have not been provided in previous studies.
We take this opportunity to thank the recommendations and suggestions made by the reviewers to improve the document.
Yours sincerely,
Authors
Reviewer 3 Report
Thanks for the effort to improve the manuscript. Nevertheless, some points should be further discussed.
- Authors said “The teeth assigned to both experimental models presented the same anatomy”. How was it verified?
- Legend of Figure 1: “(B) reconstruction if the CBCT scan”. Please, try to reformulate the sentence.
- “The interradicular spaces where the orthodontic micro-screws were placed were also randomly selected (Epidat 4.1, Galicia, Spain)”. Why did you choose a random selection?
- The strengths and limitations of the methodology, as well as clinical implications of the results obtained have not been discussed.
Author Response
Dear Reviewer 3,
I’m pleased to resubmit the manuscript of the work entitled, “Influence of the Computer-Aided Static Navigation Technique on the Accuracy of the Orthodontic Micro-screws Placement. An in Vitro Study”.
Reviewer 3: Moderate English changes required
Response: In order to adapt to the reviewer's 3 comments, we send the manuscript to a specialized English Editing Service.
Reviewer 3: Authors said “The teeth assigned to both experimental models presented the same anatomy”. How was it verified?
Response: In order to adapt to the reviewer's 3 comments, we have re-write the sentence to clarify that the anatomical dimensions of the teeth were checked with an electronic calliper.
Reviewer 3: Legend of Figure 1: “(B) reconstruction if the CBCT scan”. Please, try to reformulate the sentence.
Response: In order to adapt to the reviewer's 3 comments, we have reformulated the sentence.
Reviewer 3: “The interradicular spaces where the orthodontic micro-screws were placed were also randomly selected (Epidat 4.1, Galicia, Spain)”. Why did you choose a random selection?
Response: In order to adapt to the reviewer's 3 comments, we clarify that there are 15 interradicular spaces between the 16 teeth presented in each experimental model; however, only 10 orthodontic micro-screws were placed in each experimental model. For this reason, the interradicular spaces where the orthodontic micro-screws were placed were randomized.
Reviewer 3: The strengths and limitations of the methodology, as well as clinical implications of the results obtained have not been discussed
Response: In order to adapt to the reviewer's 3 comments, we added the strengths and limitations in the Discussion section.
Reviewer 3: The positioning of micro-screws should be explained more in detail (in correspondence of which teeth have been placed the micro-screws and why?)
Response: In order to adapt to the reviewer's 3 comments, we clarify that we have newly assessed the sample size in base of the results of the previous study, to obtain statistically significant differences: to achieve a power of 80.00% to detect differences in the contrast of the null hypothesis H₀: μ₁ = μ₂ by means of a bilateral Student's T-test for two independent samples, taking into account that the significance level is 5.00% , and assuming that the mean of the Reference group is 1.67 mm, the mean of the Experimental group is 1.25 mm and the standard deviation of both groups is 0.87 mm, it will be necessary to include 69 experimental units in the Reference group and 69 units in the Experimental group, totalling 138 experimental units in the study; which is higher than Qiu et al study (n = 30), Liu et al (n = 34), Miyazawa et al (n = 44) and Bae et al (n = 45).
Additionally, the results were assessed regarding the dental sector that was placed (incisive-canine, premolar and molar) to provide novel and useful information that have not been provided in previous studies. Moreover, depending on the treatment planning, micro-screws can be placed anywhere on the alveolar bone in order to reinforce anchorage. Keeping that in mind, we planned a study placing micro-screws between the roots of adjacent teeth in the whole arc. In the results, we divided the areas in order to study separately the differences in the micro-screws placement depending on the distance between the adjacent roots and their morphology. This can be helpful in order to provide information on the risks of micro-screws placement in each area, which has a great clinical significance.
We take this opportunity to thank the recommendations and suggestions made by the reviewers to improve the document.
Yours sincerely,
Authors
This manuscript is a resubmission of an earlier submission. The following is a list of the peer review reports and author responses from that submission.